# The Antioxidant Activities In Vitro and In Vivo and Extraction Conditions Optimization of Defatted Walnut Kernel Extract

**DOI:** 10.3390/foods12183417

**Published:** 2023-09-14

**Authors:** Xiaomei Zhou, Xiaojian Gong, Xu Li, Ning An, Jiefang He, Xin Zhou, Chao Zhao

**Affiliations:** 1Key Laboratory for Information System of Mountainous Areas and Protection of Ecological Environment, Guizhou Normal University, Guiyang 550001, China; 2Guizhou Engineering Laboratory for Quality Control & Evaluation Technology of Medicine, Guizhou Normal University, Guiyang 550001, China; 3Experimental Centre of Tropical Forestry, Chinese Academy of Forestry, Pingxiang 532600, China

**Keywords:** walnut kernel, antioxidant activity, in vitro, in vivo, response surface methodology

## Abstract

The objective of this study was to determine the antioxidant activities of defatted walnut kernel extract (DWE) and whole walnut kernel extract (WE) in vitro and in vivo. Three spectrophotometric methods, DPPH, ABTS, and FRAP, were used in in vitro experiments, and mice were used in in vivo experiments. In addition, response surface methodology (RSM) was used to optimize reflux-assisted ethanol extraction of DWE for maximum antioxidant activity and total phenolic content. The results of in vitro experiments showed that both extracts showed antioxidant activity; however, the antioxidant activity of DWE was higher than that of WE. Both extracts improved the mice’s oxidative damage status in in vivo studies. An ethanol concentration of 58%, an extraction temperature of 48 °C, and an extraction time of 77 min were the ideal parameters for reflux-assisted ethanol extraction of DWE. The results may provide useful information for further applications of defatted walnut kernels and the development of functional foods.

## 1. Introduction

Originally from the Mediterranean Basin, walnut (*Juglans regia* L.) is a member of the Juglandaceae family. The major walnut producers in the world are China and the United States [1]. Walnut is one of the four most popular nuts in the world [2,3], favored for their nutritional and medicinal value. Walnut is widely distributed in various provinces of China, such as Heilongjiang, Hebei, Yunnan, and Guizhou. China has the largest planting area and production. Data shows that China’s walnut planting area and production reached 631,000 hectares and more than 2.5 million tons in 2019, respectively [4]. Walnut kernels are rich in nutrients, containing large amounts of protein, dietary fiber, carbohydrates, vitamins, fat, and amino acids, as well as trace elements such as potassium, iron, and calcium [5,6,7]. Walnut kernels have anti-inflammatory and antiseptic effects, protecting blood vessels, lowering blood pressure, and preventing cancer. Walnut kernels can be eaten directly or processed into various food items for consumption, such as beverages, desserts, and snacks. The kernels, husks, green barks of walnuts, branches, and leaves of walnut trees are all treasures that have been extensively employed in medical research and cosmetics manufacturing [8].

The main cause of oxidative stress is inequality between prooxidants and antioxidants owing to the overproduction of reactive oxygen species (ROS) that exceeds the ability of cellular defenses [9,10]. The by-products of aerobic metabolism can damage the structure and function of cellular organelles. For example, reactive oxygen species (ROS), if not effectively removed, may harm proteins, lipids, and deoxyribonucleic acid (DNA) in cells, ultimately leading to cellular damage and cell death [11,12]. A variety of human diseases have been observed to have a bearing on oxidative stress, e.g., cancer [13], dementia [14], cardiovascular diseases, neurodegenerative diseases [15,16], hyponatremia [17], diabetes [18], uterine fibroid [19], and colorectal cancer [20]. Organisms contain natural defense systems against oxidative stress, including enzymatic antioxidant and non-enzymatic antioxidant systems. In addition, supplementation with exogenous antioxidants is also an efficacious way to effectively inhibit oxidative damage. Antioxidants can inhibit oxidative damage by scavenging or neutralizing free radicals, enhancing the activity of antioxidant enzymes in vivo, and reducing the products of lipid peroxidation [21,22]. When present in the medium in lower concentrations than oxidizable substrates, antioxidants prevent the oxidation of substrates [23,24]. In the last century, various synthetic antioxidants produced by chemical processes have been used in foods to prevent oxidation, such as butylhydroxytoluene (BHT) and butylhydroxyanisole (BHA). However, researchers have indicated that synthetic antioxidants can have negative consequences, like carcinogenesis [25,26,27]. Therefore, the search for natural alternatives from plants has generated great interest based on safety and health.

The presence of polyphenols, tocopherols, squalene, unsaturated fatty acids, phytosterols, and bioactive peptides in walnut kernels contributes to their nutritional and medicinal value [28,29]. Polyphenols, which have both antibacterial and antioxidant properties [30,31], are plentiful secondary metabolites in plants. The hydroxyl groups of polyphenolic compounds can react with free radicals, thus slowing down the harm of free radicals to cells and achieving antioxidant effects. Walnut kernels have the highest phenol content compared to other nuts, like pistachios, Brazil nuts, almonds, and peanuts [32,33]. A study even showed that walnut kernels have the greatest antioxidant ability among twenty-five common foods [34]. Therefore, walnut kernels are a good source of natural antioxidants.

Defatted walnut kernels are a by-product left over from the extraction of walnut oil. Defatted walnut kernels are rich in phenolics, which have antioxidant properties [35]. This property makes it a potential source for the development of functional foods and dietary products. The addition of substances with healthy properties can transform these products into healthier foods. Previous studies on the antioxidant activity of defatted walnut kernel have mainly focused on in vitro or in vivo studies [36,37,38]. However, many substances may not have consistent antioxidant activities both in vitro and in vivo. There are no reports on the use of in vitro and in vivo assays to comprehensively assess the antioxidant activity of defatted walnut kernel. One study used ultrasonic-assisted ethanol extraction of defatted walnut kernel extract to obtain optimal technical parameters for de-penalization treatments [39]. However, there are few studies on the optimization of defatted walnut kernel extract (DWE) based on the antioxidant effect as an indicator. Therefore, in the present study, the antioxidant activity of defatted walnut kernel extract and whole walnut kernel extract (WE) was comprehensively evaluated using in vitro and in vivo assays. In addition, response surface methodology was used to optimize extraction conditions for reflux-assisted ethanol extraction of defatted walnut kernel extracts. In this study, three spectrophotometric methods, DPPH, ABTS, and FRAP, were employed to measure the ability to eliminate free radicals in vitro. In in vivo experiments, malondialdehyde (MDA) content, total antioxidant activity (T-AOC), glutathione peroxidase (GSH-Px) activity, catalase (CAT) activity, and total superoxide dismutase (T-SOD) activity were determined in serum and tissues of mice. 

## 2. Materials and Methods

### 2.1. Materials and Reagents

This test uses walnut kernels as the raw material from Shijiazhuang City, Hebei Province. BHA and ABTS were obtained from J&K Scientific (Beijing, China). Trolox was provided by Aldrich Co. (St. Louis, MO, USA). DPPH was supplied by TCI Development Co., Ltd. (Shanghai, China). Tripyridyltriazine and D-galactose were bought from Aladdin Co., Ltd. (Shanghai, China). The malondialdehyde (MDA), total antioxidant activity (T-AOC), total superoxide dismutase (T-SOD), glutathione peroxidase (GSH-Px), and catalase (CAT) kits were provided by Nanjing Jiancheng Bioengineering Company (Nanjing, China). Other solvents utilized in the experiments were analytical- or HPLC-grade.

### 2.2. Preparation of Walnut Kernel Extracts

Weigh 0.30 g of the defatted walnut kernel (over 20 mesh) and walnut kernel dissolved in 70% ethanol at a ratio of 30:1. After being sonicated (HU-10260B, Tianjin Hengao Technology Development Co., Tianjin, China) for 40 min; the solution was centrifuged (TDL-5A, Changzhou Wanhe Instrument Manufacturing Co., Changzhou, China) to separate the supernatant, and then diluted with a 1.5% sodium carboxymethylcellulose solution to the desired concentration. Defatted walnut kernel extract (DWE) and walnut kernel extract (WE) were obtained.

The mixed fatty acids were prepared by saponifying walnut oil at 60 °C for 1 h. The saponified solution was then acidified with hydrochloric acid to pH 3–4, followed by diluting the acidified solution with anhydrous ethanol to the appropriate concentration.

### 2.3. In Vitro Antioxidant Activity Determination

Three complementary and common methods, DPPH, ABTS, and FRAP, were selected to determine the in vitro antioxidant activity of each walnut extract separately. As positive controls, the experiment employed vitamin C and butylhydroxyanisole (BHA).

#### 2.3.1. DPPH Assay

The determination of DPPH free radical elimination activity was performed according to the literature with some modifications [40]. In total, 0.5 mL of 250 mmol/L DPPH (dissolved in anhydrous ethanol) working solution was added to 0.5 mL of the appropriate concentration of sample solutions or Trolox solutions. The mixture was reacted at room temperature and protected from light for 40 min. The absorbance values were subsequently determined at 517 nm using an enzyme marker (Spectra Max plus 383, MDC, Silicon Valley, CA, USA). A series of concentrations of Trolox (3.12–100 μmol/L) solutions were utilized as standard, and the calibration curve was plotted to compute DPPH radical scavenging activity. As follows: Y_1_ = 0.008X_1_ + 0.1225 (R^2^ = 0.9982) (X_1_, concentration of Trolox, μg/mL; Y_1_, DPPH radical scavenging rate, %). DPPH scavenging activity can be shown as Trolox equivalent antioxidant capacity (TEAC).

#### 2.3.2. ABTS Assay

The determination of ABTS free radical elimination activity was performed according to the literature with some modifications [41]. For the ABTS stock solution, ABTS solution and potassium persulfate solution were produced to final concentrations of 7 mmol/L and 2.45 mmol/L, individually. After being combined, the mixing solution was left at room temperature and shielded from light over 12 h. Anhydrous ethanol was used to dilute the stock solution into the working solution until the absorbance value at 734 nm was 0.80 ± 0.02. A total of 1.0 mL of ABTS working solution was mixed with 50 μL of proper sample concentration or Trolox solutions. The mixture was reacted at room temperature and protected from light for 20 min, and the absorbance values were determined at 734 nm using an enzyme marker (Spectra Max plus 383, MDC, Silicon Valley, CA, USA). A series of concentrations of Trolox (25–800 μmol/L) solutions were utilized as standard, and the calibration curve was drafted to compute ABTS radical scavenging activity as follows: Y_2_ = 0.0012X_2_ − 0.0196 (R^2^ = 0.9954) (X_2_, concentration of Trolox, μg/mL; Y_2_, ABTS radical scavenging rate, %). ABTS scavenging activity can be shown as Trolox equivalent antioxidant capacity (TEAC).

#### 2.3.3. FRAP Assay

The determination of ferric-reducing antioxidant power was performed according to the literature with some modifications [42]. Ferric chloride solution (20 mmol/L), acetate buffer solution (pH 3.6, 0.3 mol/L), and TPTZ stock solution (10 mmol/L) were mixed in a volume ratio of 1:1:10. This resulted in a fresh FRAP working solution. The samples were prepared with anhydrous ethanol to the appropriate concentration. In total, 50 μL of the sample solutions or Trolox solutions and 1 mL of FRAP reagent were well mixed at 37 °C. After 10 min, the absorbance values were measured at 593 nm using an enzyme marker (Spectra Max plus 383, MDC, Silicon Valley, CA, USA). The calibration curve was used to compute the FRAP with Trolox as the standard: Y_3_ = 0.0011X_3_ − 0.1113 (R^2^ = 0.9996) (X_3_, concentration of Trolox, μg/mL; Y_3_, FRAP). FRAP was signified as Trolox equivalent antioxidant capacity (TEAC).

### 2.4. In Vivo Antioxidant Activity Determination

#### 2.4.1. Animals

Six to eight-week-old Kunming mice (SPF grade, Batch No. SCXK 2009-0012), weighing roughly 20 ± 2 g, were obtained from Changsha Tianqin Biotechnology Company (Changsha, China). The feed was purchased at Changsha Tianqin Biotechnology Company. Before the start of the experiment, the mice were fed with basic feed and water for one week to acclimatize them to the conditions of the animal room. Animal experiments in this study were conducted in accordance with current national and international laws and recommendations, and every effort was made to minimize suffering. The experiment was approved by the Animal Care and Use Committee of Guizhou Normal University (C00032801).

#### 2.4.2. Mice Grouping and Experimental Design

After the mice were adapted, fifty mice were randomly divided into five groups of ten mice each as follows: the normal control group (Group I), the model control group (Group II), the positive control group (Group III); the defatted walnut kernel extract (DWE) group (Group IV), and the whole walnut kernel extract (WE) group (Group V). Vitamin C was the positive control that was employed.

The doses administered to the WE and DWE groups were set based on the defatted walnut powder extract group. To produce an oxidative damage model, mice in Groups II through V received an intraperitoneal injection of 0.3 mL of D-galactose solution (0.5 g/kg/d). Mice in Group I received a 0.3 mL physiological saline injection. In addition, Vitamin C (0.2 g/kg/d), DWE (0.2 g/kg/d), and WE (0.5 g/kg/d) were given daily to groups III, IV, and V, respectively. This was conducted for 42 consecutive days by gavage.

After the last gavage, all groups of mice were given a fast while still having access to water. The next day, blood was taken from the eyeballs of mice. Subsequently, after standing for a while, the collected blood was centrifuged (4 °C, 4000 rpm, 10 min) (TGL-16M, Changsha Meijiasen Instrument Co., Changsha, China) to collect the serum from the supernatant. The brain tissue, kidney, liver, and heart of mice were immediately dissected. The blood on the tissues was washed away separately with ice-cold physiologic saline and blotted dry with filter paper. The homogenization process involved combining the sample with iced saline at a ratio of 1:9 (mass to volume) to obtain a 10% homogenate. The resulting mixture was then subjected to centrifugation at 4 °C and 4000 revolutions per minute (rpm) for 5 min using a TGL-16M centrifuge (Changsha Meijiasen Instrument Co., Changsha, China). The supernatant was collected and preserved at a temperature of −20 °C for subsequent experiments.

#### 2.4.3. Antioxidant Assays

The Technical Specification for Inspection and Evaluation of Health Food (2003 edition) stipulates that when any of the indicators of lipid peroxide content and any of the indicators of antioxidant function enzyme activity are positive, it can indicate that the tested extract has antioxidant health function. Therefore, we made a mouse model of oxidative damage using D-galactose. Subsequently, measurements of total antioxidant activity (T-AOC), catalase (CAT) activity, total superoxide dismutase (T-SOD) activity, glutathione peroxidase (GSH-PX) activity, and malondialdehyde (MDA) content were performed in the serum and tissues of mice. The operation procedure was regulated on the basis of the operating descriptions of the kit. 

### 2.5. Optimization of Extraction Conditions of Defatted Walnut Kernel Extract

The extraction conditions for reflux-assisted ethanol extraction of defatted walnut kernel extract were optimized for maximum antioxidant activity and total phenolic content using single-factor experiments and response surface methodology.

#### 2.5.1. Single-Factor Experimental Design

In this study, 0.30 g of defatted walnut powder was weighed. Firstly, the best extraction method was determined by a single-factor experiment as reflux extraction. The best extraction solvent was anhydrous ethanol, and the best liquid-to-material ratio was 30:1 (mL/g). Based on this, further single-factor experiments were carried out to determine the optimal extraction conditions for defatted walnut kernel extract (DWE). The extraction conditions included extraction time, ethanol concentration, and extraction temperature. Briefly, the extraction times were 40 min, 60 min, 80 min, 100 min, 120 min, and 140 min, respectively. The ethanol concentrations were 0%, 20%, 40%, 60%, 80%, and 100%, respectively. The extraction temperatures were 35 °C, 45 °C, 55 °C, 65 °C, and 75 °C, respectively. The common conditions were an extraction time of 80 min, an ethanol concentration of 50%, and an extraction temperature of 60 °C. When one of the independent variables is modified, the other parameters remain unchanged. The antioxidant activity was determined according to Section 2.3.1, Section 2.3.2 and Section 2.3.3. The total phenol content (TPC) was performed according to the literature with some modifications [43].

Add 2.5 mL of Folin–Ciocalteu phenol reagent (10%) to 0.1 mL of the appropriate concentration of DWE or 100 μg/mL gallic acid standard solution. Allow to stand for 5 min. Then, add Na_2_CO_3_ solution (10%, 2 mL) and distilled water to bring the volume to 10 mL. After another hour of keeping the mixture at room temperature and shielded from light, the absorbance values were gauged at 765 nm. The TPC of the samples was represented as gallic acid equivalents per gram of dry weight.

#### 2.5.2. Response Surface Methodology (RSM) Experimental Design

Following the results of the single-factor experiments, the defatted walnut kernel extract (DWE) was optimized by response surface methodology (RSM) using Central Composite Design (CCD). Ethanol concentration, extraction time, and extraction temperature were the independent variables, and ABTS, DPPH, FRAP, and total phenol content (TPC) were the response variables. Five distinct levels (−1.68, −1, 0, 1, 1.68) of study were performed on each design variable (Table 1). A total of 19 randomized experimental cycles were run. Table 2 displays the response surface methodology run design and outcomes. Following the completion of the CDD matrix, an analysis of variance (ANOVA) was conducted, and the data were fitted to a second-order polynomial model.
$$y=\beta 0+\sum_{i=1}^{k}\beta ixi+\sum_{i=1}^{k}\beta iixi2+\sum_{i}^{k-1}\sum_{j}^{k}\beta ijxixj$$
where *y* denotes the response variable, *xi* and *xj* represent the independent variables, and *k* is the number of independent variables (*k* is 3 in this study). *β*0, *βi*, *βii*, and *βij* indicate the intercept term coefficient, the linear term coefficient, the quadratic term coefficient, and the interaction term coefficient, respectively.

### 2.6. Statistical Analysis

All experiments in vivo were performed in eight replicates and averaged. All other experiments were repeated three times and averaged. The experimental result values are expressed as mean ± SD. The data obtained from in vivo and in vitro experiments were statistically analyzed by SPSS.18.0 software. A one-way analysis of variance (ANOVA) was used, followed by Student’s test to assess significance. Statistically, significant results were signified as *p* < 0.05, more significant as *p* < 0.01, and extremely significant as *p* < 0.001. The optimization experiments were statistically analyzed using Design-Expert 8.0.6 software.

## 3. Results and Discussion

### 3.1. In Vitro Antioxidant Activities of Walnut Kernel Extracts

Table 3 summarizes the antioxidant activities of walnut kernel extracts determined by DPPH, ABTS, and FRAP methods. Vitamin C and butylhydroxyanisole (BHA) were employed as positive controls. Walnut kernels contain up to 65–70% oil, while fatty acids make up more than 90% of walnut oil. Therefore, the antioxidant potential of fatty acids was evaluated and compared with defatted walnut kernel extract (DWE) and whole walnut kernel extract (WE). The results suggested that extracts from all parts of walnut kernels had certain in vitro antioxidant effects. After oil removal, the cake meal extract of walnut kernels had the strongest antioxidant capacity in vitro. Moreover, trends in results measured by the three methods were consistent, with vitamin C > BHA > DWE > WE > fatty acids. Specifically, the results of DPPH, ABTS, and FRAP in DWE (0.025 mg/mL) were 1026.19 ± 52.95 μM TEAC/g, 2754.30 ± 42.79 μM TEAC/g, and 5829.09 ± 36.36 μM TEAC/g. The clearance rate measured by DPPH and ABTS at a concentration of 0.025 mg/mL of WE was small and not in the linear range. In contrast, the FRAP result was 4180.61 ± 41.99 μM TEAC/g. At a fatty acid concentration of 100 mg/mL, the clearance measured by the three methods was small and not in the linear range. This showed that DWE has significantly different (*p* < 0.01) antioxidant activity from WE and fatty acids. Compared to BHA, vitamin C showed the greatest antioxidant activity among the two positive controls. The result measured by the DPPH method was 4006.05 ± 4.03 μM TEAC/g. The result measured by the ABTS method was 9018.41 ± 85.58 μM TEAC/g. The result measured by the FRAP method was 20,018.69 ± 116.89 μM TEAC/g. Interestingly, the antioxidant activities of walnut kernel extracts showed high differences in DPPH, FRAP, and ABTS. Under the same conditions, the FRAP method showed high antioxidant capacity and large differences. For example, for the antioxidant-active substance DWE, the antioxidant activity determined through the FRAP method increased 5.68-fold and 2.12-fold compared to DPPH and ABTS, respectively. This result is due to the different detection principles of the methods utilized. Each method has its range of applications and features. At present, no method can comprehensively measure the antioxidant activity of a substance. Consequently, the antioxidant activities of food extracts should be evaluated using various methodologies simultaneously in experiments since different approaches frequently provide different findings [44].

Oxidative damage is an important cause of aging in organisms. Antioxidant and anti-aging are closely related. Plant extracts are rich in bioactive substances. These bioactive substances are highly valuable in antioxidants and disease prevention. Therefore, antioxidant studies on plant extracts can provide many possibilities for the development of natural antioxidant products.

Walnut kernels are one of the most beloved nuts in the world, and they are widely grown for this reason. In addition to their good taste, walnuts have great benefits for human health. The reason for this is that walnut kernels are rich in natural active substances. Phenolic compounds are one of these naturally active substances. Phenolic compounds have been shown to have antioxidant properties [45,46,47]. Walnut kernels have a higher antioxidant potential than other nuts due to their high phenolic content. Defatted walnut kernels, a by-product released after obtaining walnut oil, also have antioxidant properties. Defatted walnuts are becoming an inexpensive source of phenolic compounds.

Our study determined the ability of defatted walnut kernel extract and whole walnut kernel extract to scavenge free radicals in vitro. The results of this study confirmed the antioxidant activity of both extracts. Previous studies have shown that most of the antioxidant activity in walnut kernels is in defatted walnut kernels [36,48,49]. Our results are consistent with them. Walnut kernels contain 65–70% oil. Regrettably, walnut kernel oil contains very low levels of phenolics, contributing less than 5% to the total antioxidant activity of walnut kernels [48]. The reason may lie in the low oil solubility of phenolic compounds [50] or be related to the distribution of phenolic compounds in walnut kernels. Defatted walnut kernels consist primarily of the skin that encases the kernel. The highest phenolic content was found in the pericarp [51,52]. Therefore, the antioxidant activity of DWE is sensibly superior to that of WE, which may also be related to the interference of walnut oil.

### 3.2. In Vivo Antioxidant Activities of Walnut Kernel Extracts

The malondialdehyde (MDA) content, the catalase (CAT) activity, total superoxide dismutase (T-SOD) activity, glutathione peroxidase (GSH-Px) activity, and total antioxidant activity (T-AOC) in serum, brain tissue, liver, kidney, and heart of mice were examined to assess the in vivo antioxidant activity of walnut kernel extracts. Figure 1 displays the results.

#### 3.2.1. Effect on Lipid Peroxidation

As shown in Figure 1A, the malondialdehyde (MDA) content in serum and all tissues was obviously higher in the model control group (Group II) than in the normal control group (Group I). It indicates that mice injected intraperitoneally with D-galactose solutions could successfully create a model of oxidative damage. In comparison to the model group, the content of MDA in the serum and tissues of mice in the vitamin C, defatted walnut kernel extract (DWE), and whole walnut kernel extract (WE) groups were reduced and varied significantly (*p* < 0.05). It showed that two walnut kernel extracts and vitamin C may both lower the MDA content of aging mice due to D-galactose to varying degrees and protect mice from lipid peroxide damage. Walnut kernel extracts showed conspicuous antioxidant effects.

#### 3.2.2. Effect on Antioxidant Enzyme Activities and Total Antioxidant Activities

As shown in Figure 1B–E, total superoxide dismutase (T-SOD) activities, glutathione peroxidase (GSH-Px) activities, catalase (CAT) activities, and total antioxidant activity (T-AOC) in the serum and various tissues were noticeably reduced in the model control group (Group II) compared with the normal control group (Group I). The ability to scavenge oxygen-free radicals in vivo was also correspondingly reduced. It indicates that intraperitoneal injection of D-galactose solution could successfully create a mouse model of oxidative damage. Figure 1B shows the effect of walnut kernel extracts on T-AOC in the serum and tissues of mice. The T-AOC in the serum and tissues of mice varied significantly among the vitamin C group, the defatted walnut kernel extract (DWE) group, and the whole walnut kernel extract (WE) group compared to the model group (*p* < 0.05). It notes that walnut kernel extracts and vitamin C could improve the T-AOC of D-galactose-aged mice to varying degrees and enhance their ability to scavenge free radicals. Figure 1C–E reflect the effects of walnut kernel extracts on antioxidant enzyme activity in the serum and tissues of mice. The activities of antioxidant enzymes in mice were reduced after the use of D-galactose. T-SOD activities, GSH-PX activities, and CAT activities in serum and tissues were improved after treatment with vitamin C, DWE, and WE. And all of them were significantly different from the model group (*p* < 0.05). Antioxidant enzymes can eliminate free radicals, maintain the redox balance of the organism, and play a certain antioxidant role in the organism.

Free radicals readily attack unsaturated fatty acids in biological membranes and cause lipid peroxidation. This action produces lipid peroxidation products, including MDA. MDA is a cytotoxic substance that can induce cross-linked polymeric of proteins, nucleic acids, and various biological macromolecules [53]. The MDA content is an indicator of the extent of lipid peroxidation occurring within the organism and may further indirectly reflect the level of cellular oxidative damage. The defense system of biological organisms includes both enzymatic and non-enzymatic antioxidant systems. T-AOC is a combination of enzymatic and non-enzymatic activity in a biological organism. The numerical value of T-AOC could objectively mirror the strength of the body’s total antioxidant activity and further indirectly reflect the body’s ability to defend against antioxidant damage. T-SOD, CAT, and GSH-Px are endogenous antioxidant enzymes that are crucial in the mechanism of antioxidant defense [54]. In the organism, these three antioxidant enzymes are the first line of defense against reactive oxygen species. T-SOD is essential for the oxidation and antioxidant balance of living organisms, converting superoxide anions into hydrogen peroxide (H_2_O_2_) and thus protecting cells from damage [55]. Hydroxyl radicals are chemically very active and can react with organic substances in living organisms, such as sugars, amino acids, organic acids, and phospholipids. The reaction speed is fast, and the damage to the cells of biological organisms is strong. However, it can be broken down into H_2_O_2_ by the highly destructive CAT under certain conditions to protect the cells of living organisms [56]. GSH-Px exclusively catalyzes the reduction reaction of reduced glutathione (GSH) to hydrogen peroxide, thus acting to defend the structure and functional completeness of the cytomembrane. These antioxidant indicators were therefore selected to assess the antioxidant activity of walnut kernel extracts.

Vitamin C is one of the most common and strong antioxidants that can perform antioxidant functions by reacting with free radicals and inhibiting lipid peroxidation damage. The selection of vitamin C as a positive control can distinctly elucidate the effect of antioxidants on antioxidant enzyme activity in oxidatively damaged mice. D-galactose is a well-stabilized reducing monosaccharide that increases tissue osmolarity and produces oxidative stress. The products of the enzyme-catalyzed reaction of D-galactose cannot be further metabolized. Instead, they accumulate in the body, damaging its antioxidant defense system and producing an overabundance of oxygen radicals. This results in a model of oxidative damage.

### 3.3. Single Factor Experimental Results

One of the primary factors impacting the total phenolic content (TPC) and antioxidant activity is the extraction variable. The results revealed that extraction time, ethanol concentration, and extraction temperature affected the antioxidant activity and TPC of defatted walnut kernel extract (DWE) (Figure 2). Specifically, the effect of extraction time on antioxidant activity manifested that the extraction time (from 40 min to 80 min) was positively correlated with antioxidant activity. The antioxidant activity, however, fell after 80 min of extraction time. Interestingly, the TPC reflects the same situation. The TPC was positively associated with the extraction time (from 40 min to 100 min). But sadly, the TPC started to progressively decline after the extraction time surpassed 100 min. Concerning the ethanol concentration, as the ethanol concentration rose from 0% to 40%, the antioxidant activity, as determined by the DPPH and ABTS techniques, increased. Subsequently, it gradually decreased. While the FRAP method showed that the antioxidant activity increased as ethanol concentration grew (from 0% to 80%), it gradually declined after ethanol concentration increased over 80%. The TPC was positively correlated with the ethanol concentration, within the range of 0–60% ethanol concentration, arriving at a maximum of 60%. After that, it gradually decreased with increasing ethanol concentrations. It is noteworthy that the extraction temperature showed the same trend as the extraction time. In general, the effects of these three variables on antioxidant activity and TPC followed a consistent trend: an early rise followed by a slow decline.

Reflux extraction is a method of extracting active ingredients from plants with volatile organic solvents. The reflux extraction method is relatively simple to operate and has a high extraction efficiency. Ethanol, acetone, methanol, and water are organic solvents often used for the extraction of phenolic compounds [57], including the extraction of walnut kernel extract. Ethanol is a hydrophilic organic solvent with good solubility for all types of chemicals. Ethanol is safer compared to methanol and acetone. During the extraction process, the performance of the extraction is affected by some external factors, such as extraction temperature, feed/liquid ratio, extraction time, and solvent concentration [58].

The variable of extraction time may have a bearing on the instability of antioxidant active components in the DWE. Additionally, when extraction time lengthens, production costs rise. Taking all aspects into consideration, the extraction time was chosen to be controlled at about 80 min. For the variable of ethanol concentration, this may be due to the fact that polyphenols are not the only antioxidant active ingredient in the DWE. The content and type of antioxidant substances extracted differed at different ethanol concentrations. From the consideration of cost and subsequent operation, it is appropriate to control the ethanol concentration at about 50%. For the extraction temperature, this may be due to the poor thermal stability of polyphenols and the destruction of active ingredients due to oxidation reactions at high temperatures. In addition, the dissolution of impurities increases at high temperatures, and the subsequent separation and purification become more difficult. Taking all factors into consideration, the extraction temperature was controlled at about 45 °C as appropriate.

### 3.4. Correlation Analysis of Antioxidant Activity and Total Phenolic Content

The content of phenolic substances contained in nuts affects their antioxidant activity [59]. According to previous studies, phenolics and antioxidant activity are noticeably positively correlated [60]. Correlations were analyzed to evaluate the relevance of total phenolic content (TPC) and antioxidant activity of walnut kernel extracts. As can be seen in Table 4, using the DPPH, ABTS, and FRAP methods, the correlation coefficients between TPC and antioxidant activity were 0.517, 0.566, and 0.539, respectively. This indicated a significant relevance in antioxidants and TPC (*p* < 0.05). The results were in accordance with previous studies. There was an extremely significant relevance (r > 0.97; *p* < 0.001) among the three antioxidant tests in defatted walnut kernel extract (DWE). Therefore, the choice of any of these methods allows a reasonable one-way analysis of variance (ANOVA) of the influencing factors. 

### 3.5. Response Surface Methodology to Optimize the DWE Extraction Conditions

#### 3.5.1. ANOVA and Quadratic Regression Analysis

Defatted walnut kernel extract (DWE) extraction conditions were optimized using the response surface method (RSM) based on the findings of single-factor experiments. The one-way analysis of variance (ANOVA) of influential variables was conducted using ABTS. As displayed in Table 5, the total phenolic content (TPC) of defatted walnut kernel extract (DWE) was more significantly correlated with extraction time (*p* < 0.01). It was extremely significantly correlated with ethanol concentration and extraction temperature (*p* < 0.001). The influencing variables were in the following order: ethanol concentration had the highest degree of influence, followed by extraction temperature and extraction time. Furthermore, the effect of extraction factors on the TPC was ranked in the following order: ethanol concentration > extraction time > extraction temperature. The quadratic multiple regression equations that were obtained to assess the ABTS and TPC outcomes are as follows: y (ABTS) = 105.25 + 9.26a + 3.33b + 6.64c − 0.20ab − 4.52ac − 0.81bc − 7.67a^2^ − 5.65b^2^ − 5.17c^2^; y (TPC) = 40.10 + 1.47a − 0.52b − 0.017c − 2.05ab − 0.36ac + 0.096bc − 2.86a^2^ − 0.43b^2^ − 0.59c^2^. It also shows that the effects of these three effects on antioxidant activity and TPC are not simply linear but quadratic. And there is an interaction between the three factors.

#### 3.5.2. Response Surface Analysis

Three-dimensional response surface plots show the interactions between each of the two factors (Figure 3). Under the constant extraction temperature, the total polyphenol content (TPC) and ABTS scavenging activity increased and then decreased with the increase in extraction time and ethanol concentration. However, the magnitude of the change in ethanol concentration was more obvious. When the extraction time remains constant, the ABTS scavenging activity and TPC rise and then decline with the rise in extraction temperature and ethanol concentration. The impact of ethanol concentration on ABTS scavenging activity and TPC, however, was more remarkable. Under the constant ethanol concentration, the TPC and ABTS scavenging activities increased and then decreased with the increase in extraction time and extraction temperature. For ABTS scavenging capacity, the magnitude of the change in extraction temperature was more obvious. For TPC, the effect of extraction time on TPC was more prominent. The analysis of variance also supports these findings.

According to response surface methodology, the ethanol concentration of 57.67%, extraction temperature of 47.81 °C, and extraction time of 76.91 min were the optimal conditions for the extraction of DWE. The ABTS scavenging activity was predicted to be 108.04 μM TEAC, and the measured value was 108.93 μM TEAC. The predicted TPC was 40.33 mg/g, and the measured value was 40.76 mg/g. Considering the convenience of operation in practical production, the ethanol concentration of 58%, extraction temperature of 48 °C, and extraction time of 77 min were determined as the optimal extraction conditions.

In the past, the amount of walnut kernels needed for the oil extraction process was almost two times greater than the amount of oil that was produced. The remaining residue, without any economic value, is used as animal feed or plant fertilizer and may even be thrown away directly as garbage. Effective utilization of these by-products not only expands the processing industry chain of walnuts but also helps to reduce the waste of resources and protect the environment. The high antioxidant activity of DWE allows it to be used as a valuable ingredient in dietary supplements, nutraceuticals, and functional products. Or as a natural preservative to reduce rapid food spoilage. It can even work synergistically with other natural antioxidant extracts to enhance their performance. Therefore, the optimization of DWE provides a reference value for the application of defatted walnut kernels.

## 4. Conclusions

This study evaluated the antioxidant activity of defatted walnut kernel extract (DWE) and whole walnut kernel extract (WE) in vitro and in vivo experiments. In vitro experiments showed that DWE had better DPPH free radical scavenging, ABTS free radical scavenging, and ferric-reducing antioxidant abilities. In vivo tests revealed that both extracts increased the activity of antioxidant enzymes and decreased lipid peroxidation products in the serum and tissues of mice. In addition, the extraction conditions for reflux-assisted ethanol extraction of DWE were effectively optimized using response surface methodology. These extraction conditions include ethanol concentration, extraction temperature, and extraction time. DWE showed maximum antioxidant activity and total phenolic content under these optimized conditions. The results confirm that defatted walnut kernel extract is a natural antioxidant and can be a potential source for the development of functional foods. It is, however, necessary to conduct further research in order to determine the effectiveness of DWE added to foods.

## Figures and Tables

**Figure 1 foods-12-03417-f001:**
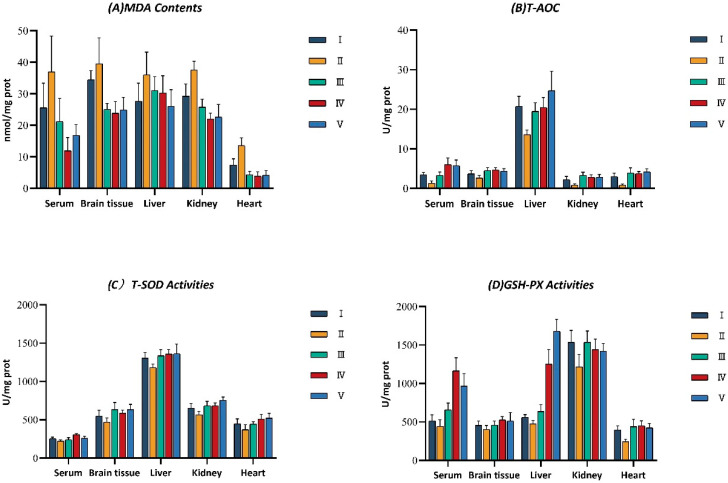
The changes of (**A**) malondialdehyde (MDA) contents; (**B**) total antioxidant activity (T-AOC); (**C**) total superoxide dismutase (T-SOD) activities; (**D**) glutathione peroxidase (GSH-Px) activities; and (**E**) catalase (CAT) activities. Data are expressed as mean ± SD (n = 8). Mice were divided into five groups as follows: I, normal control group (saline); II, model control group (D-gal solution); III, positive control group (VC solution, 0.2 g/kg/d); IV, DWE (walnut kernels extract, 0.2 g/kg/d); V, WE (walnut kernels extract, 0.5 g/kg/d). Compared with the model group, *p* < 0.05.

**Figure 2 foods-12-03417-f002:**
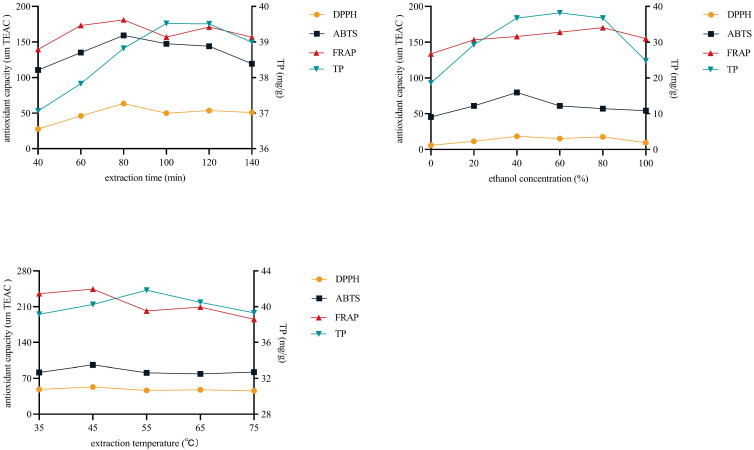
Effect of extraction variables on antioxidant activity (DPPH, ABTS, FRAP) and total phenolic content (TP) of defatted walnut kernel extract (DWE). Min, minutes.

**Figure 3 foods-12-03417-f003:**
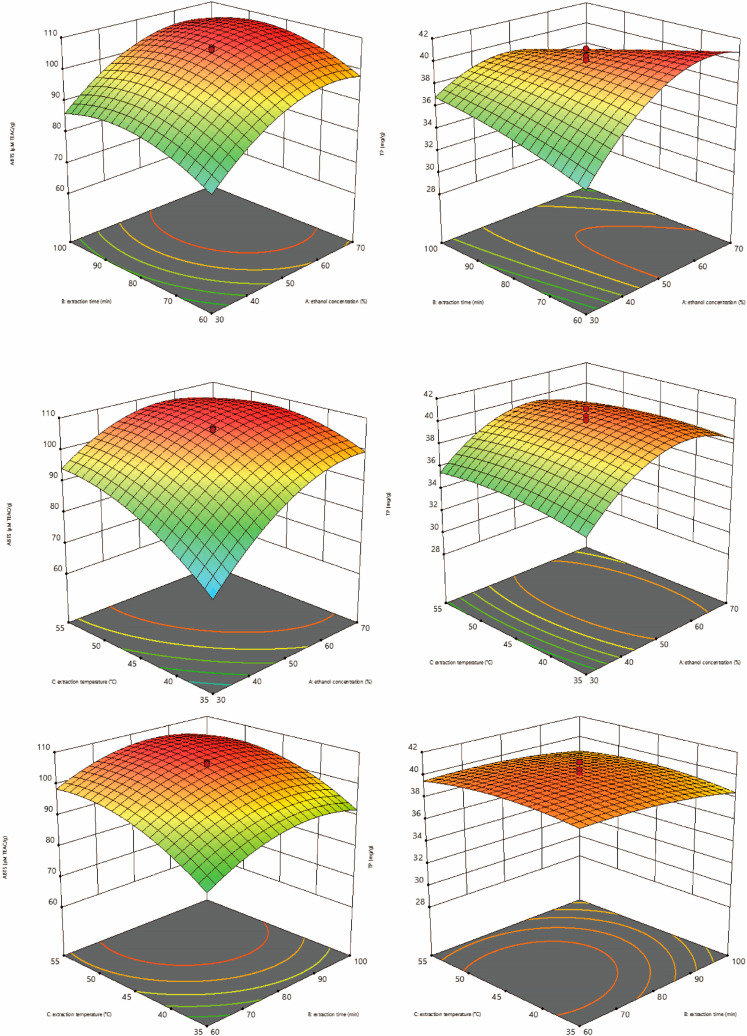
Response surface plots of antioxidant activity (ABTS) and total phenolic content (TP) of defatted walnut kernel extract (DWE) as influenced by the interaction of paired factors.

**Table 1 foods-12-03417-t001:** Independent variables, real and coded values in Central Composite Design (CCD).

Variable	Levels				
	−1.68	−1	0	1	1.68
Ethanol concentration (A, %)	16.36	30	50	70	83.64
Extraction time (B, minutes)	46.36	60	80	100	113.64
Extraction temperature (C, °C)	28.18	35	45	55	61.82

**Table 2 foods-12-03417-t002:** Central Composite Design (CCD) scheme and experimentally observed antioxidant activity (ABTS, DPPH, FRAP) and total phenolic content (TPC) response values.

Run	Variables			Response Values			
	A	B	C	ABTS	DPPH	FRAP	TPC
1	30	60	55	85.91	37.04	220.79	32.84
2	50	80	45	106.98	46.52	247.49	41.13
3	50	80	45	106.32	46.37	250.13	40.13
4	50	46.36	45	81.05	35.70	215.33	39.53
5	50	80	28.18	75.79	31.49	199.88	39.04
6	16.36	80	45	67.25	29.03	192.61	29.83
7	50	113.64	45	92.66	40.31	225.33	40.66
8	83.64	80	45	95.04	41.37	225.94	36.58
9	50	80	61.82	100.63	42.37	232.61	40.23
10	50	80	45	103.50	43.98	252.70	40.48
11	70	60	35	92.43	39.43	224.12	39.87
12	70	60	55	98.34	42.25	232.30	39.38
13	30	60	35	64.17	28.56	180.18	33.85
14	70	100	55	101.71	42.65	243.21	32.25
15	70	100	35	101.27	42.51	234.73	34.32
16	50	80	45	103.72	43.52	252.59	38.52
17	30	100	35	71.57	29.83	191.70	34.54
18	50	80	45	106.56	46.11	247.44	39.83
19	30	100	55	92.32	38.17	221.09	35.88

A, ethanol concentration; B, extraction time; C, extraction temperature; ABTS, ABTS radical scavenging activity of defatted walnut kernel extracts; DPPH, DPPH radical scavenging activity of defatted walnut kernel extracts; FRAP, ferric-reducing antioxidant power of defatted walnut kernel extracts; TPC, total phenolic content of defatted walnut kernel extracts.

**Table 3 foods-12-03417-t003:** Comparison of antioxidant activities of different parts in walnut kernels.

Sample	Concentration (mg/mL)	DPPH (μM TEAC/g)	ABTS (μM TEAC/g)	FRAP (μM TEAC/g)
Vitamin C	0.025	4006.05 ± 4.03	9018.41 ± 85.58	20,018.69 ± 116.89
BHA	0.025	3660.22 ± 13.98	7301.33 ± 171.15	16,362.42 ± 575.34
DWE	0.025	1026.19 ± 52.95	2754.30 ± 42.79	5829.09 ± 36.36
WE	0.025	ND	ND	4180.61 ± 41.99 ^a^
Fatty acid	100	ND	ND	ND

BHA: butylhydroxyanisole; DWE: defatted walnut kernel extract; WE: whole walnut kernel extract. Data are expressed as mean ± SD (n = 3). Superscript letters indicate a more significant difference (*p* < 0.01) compared to DWE. ND, not detected.

**Table 4 foods-12-03417-t004:** Correlations between total phenolic content (TPC) and antioxidant activities (DPPH, ABTS, FRAP).

	FRAP	DPPH	ABTS
**DPPH**	0.971 **	1	–
**ABTS**	0.976 **	0.991 **	1
**TPC**	0.517 *	0.566 *	0.539 *

** *p* < 0.01; * *p* < 0.05.

**Table 5 foods-12-03417-t005:** ANOVA for CCD design.

Source	Sum of Squares	df	Mean Square	F-Value	*p*-Value	Significance
**ABTS Model**	3352.62	9	372.51	38.43	<0.0001	***
**A**	1172.04	1	1172.04	120.92	<0.0001	***
**B**	151.87	1	151.87	15.67	0.0033	**
**C**	601.23	1	601.23	62.03	<0.0001	***
**AB**	0.32	1	0.32	0.033	0.8597	
**AC**	163.12	1	163.12	16.83	0.0027	**
**BC**	5.22	1	5.22	0.54	0.4815	
**A^2^**	803.14	1	803.14	82.86	<0.0001	***
**B^2^**	435.69	1	435.69	44.95	<0.0001	***
**C^2^**	364.90	1	364.90	37.65	0.0002	***
**Residual**	87.23	9	9.69			
**Lack of Fit**	76.09	5	15.22	5.46	0.0625	Not significant
**Pure Error** **R^2^**	11.14 0.9746	4	2.79			
**TPC**						
**Model**	180.26	9	20.23	5.75	0.0078	**
**A**	29.47	1	29.47	8.46	0.0174	*
**B**	3.64	1	3.64	1.04	0.3334	
**C**	3.83 × 10^−3^	1	3.83 × 10^−3^	1.10 × 10^−3^	0.9743	
**AB**	33.66	1	33.66	9.66	0.0125	*
**AC**	1.04	1	1.04	0.30	0.5974	
**BC**	0.074	1	0.074	0.021	0.8872	
**A^2^**	111.96	1	111.96	32.14	0.0003	***
**B^2^**	2.50	1	2.50	0.72	0.4189	
**C^2^**	4.76	1	4.76	1.37	0.2724	
**Residual**	31.35	9	3.48			
**Lack of Fit**	27.61	5	5.52	5.90	0.0550	Not significant
**Pure Error** **R^2^**	3.74 0.8518	4	0.94			

TPC, total phenolic content; A, ethanol concentration; B, extraction time; C, extraction temperature. NS, not significant; *, significant (*p* < 0.05); **, more significant (*p* < 0.01); ***, extremely significant (*p* < 0.001).

## Data Availability

The data used to support the findings of this study can be made available by the corresponding author upon request.

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
