# Peer review of "The Antioxidant Activities In Vitro and In Vivo and Extraction Conditions Optimization of Defatted Walnut Kernel Extract"

_foods, 2023, doi:10.3390/foods12183417_

Round 1

Reviewer 1 Report

Manuscript title: The antioxidant activities in vitro and in vivo and extraction conditions optimization of defatted walnut kernel extract

This study has certain significance in the antioxidant activities of defatted walnut kernel ... However, revisions are necessary for the current version of the manuscript. The following questions to be addressed/considered may be helpful to improve the manuscript.

Major comments

·         Insufficient Abstract: In the abstract, the main aim and background of the manuscript are missing, the current version it only highlights the result. In addition, it would be even better to have a sentence as a future perspective.

·         The unit/abbreviation is not mentioned before, consider defining the abbreviation when mentioned for the first time…. Please check throughout the manuscript to define the abbreviations.

·         Lake of scientific literature to support the statements and findings throughout the manuscript…... I have made some suggestions for that and more need it….

·         More information is needed for ALL TABLE captions and define the abbreviation and units that are used. And adjust the significant figures for the table and manuscript.

·         ·         I have a major concern about the results and discussion section. The authors describe the results and compare the results with previous studies, however, insight mechanisms are still insufficient.

Specific comments:

Introduction:

Line 31-34: A complicated sentence, please revise and check the grammar

Line 38: A reference is needed here:

Line 38-39; a very descriptive sentence, please delete.

Line 71: A reference is needed here, for example, you can use:

https://doi.org/10.1080/22297928.2016.1152912

Line 74-78: A complicated sentence, please revise and check the grammar

line 81: A reference is needed here, for example, you can use:

https://doi.org/10.1021/acs.jafc.9b07160

line 84: A reference is needed here

In MM section

Literature references are missing for all sub-section. It would be better to cite the references that the procedure adopted.

Additional info is needed for the table caption, most importantly significant figures.

In MM section, what is the quality control (QC) data? There is no mention of the QC.

What is the accuracy of the instruments, recovery, LOD, and LOQ ……. These parameters are needed to report the efficiency of any analytical system.

In general, how many times you’ve recorded the data,? duplicate? Triplicate?..... what you mentioned in the text is not clear, please elaborate more on this

2.5. Statistical Analysis

How the comparison was made between the treatments? Ad see my comment for Figures

R&D section

These sections repeat information already presented and explain things in an unnecessarily complicated way. The quality of the manuscript would benefit from the whole section being condensed, Line 280-305, Line 344-368, Line 442-456, and Line 472—497…..

Figure 1. How the comparison is made between the organs and for the statistical analysis how we can wee the significant changes here?

Line 206-208:  I am not sure how Yao et al. related to your data. Please provide another reference.

Conclusion

I believe there are other important conclusions that could be made from this study…. And the future perspectives for the following research are highly crucial here.

Important conclusions! However, the future perspectives for the following research are highly crucial here …..

The section should not be a summary of your study or an extension of the discussion. This section should illustrate the mechanistic links of findings of this study. The conclusions should answer the hypothesis of your study and should focus on the implication of your findings. Remember that the conclusions must be self-explanatory. This section should still highlight the novelty and implication of your study also.

Grammar and punctuation issuers need to be addressed. I have selected/mentioned some as examples.

Reviewer 2 Report

In this work, the extract of defatted walnut kernel was prepared, and the antioxidant activity of this extract was examined in vitro and in vivo. Further study was to optimize the ethanol reflux extraction of DWE using response surface methodology (RSM).

 Abstract section seemed well written, and had nothing worthy of special mention. Several small points would probably be better to change “the ability” (line 15) and “antioxidant activity” (line 16) to “their abilities” and “their antioxidant activities”. The word “activity” (line20) should be insufficient and inadequate, and easy to misunderstand. It might be better a little to say “activities”. Also, it could not be determined whether the words “activity” and “capacity” were possibly correct to use equally. In other words, these words would have the same meaning, and could be use the same way.  

1. Introduction section would probably be able to provide enough information regarding the background and the rationale of this work.

2. Materials and Methods section: The phrases “analysis-level” and “HPLC-level” (lines 107-108) were incorrect chemical terminology. First, the word “level” was inadequate, and it should be “grade”. Second, the word “analysis” seemed wrong, and it should be “analytical”. Therefore, these phrases should be “analytical- or HPLC-grade”. Besides, according to the contents, it would probably be better to replace the word “Acquisition” (line 109) with the other “Preparation”. The word “respectively” should be redundant and unnecessary, and it should be deleted. The word “gauged” (line 126)

seemed not popular and not in common use, and it would probably be better to say “measured” or “determined”. The word “criteria” (line 127) might be the same, and it should be “standard”.  

 In many places, there were unnatural and inadequate words observed, and they were rarely used and should be better to rephrase them. Also, there were many words seemed not necessary or improper positions, for example “concentration” (line 138), “formulated” (line 148), “mingled” (line 150). The phrase “6–8 weeks old mice” should be 6–8 week–old mice”. In general, the explanation of the experimental conditions, particularly the animal experiments, seemed quite awkward and poor. For example, the phrase “in the form of mean ± SD” (line 238) seemed quite unnatural.

 3. Results and Discussion section might have small grammatical errors, such as the misuse of singular or plural forms The word “activity” (lines 244 and 245) would probably be a plural form “activities”. Also, the phrase “the results of” (line 245) might not be necessary and should be deleted. Also, the word “activity” (line 245) should be a plural form “activities”. The phrase “account for” (line247) seemed unnatural and strange, and it would probably be a passive form or might be better to use other verbs. Just like these, there were many small mistakes observed, and they should be appropriately revised.

 With regard to scientific interest and significance, this work could be considered not so highly valuable and important, but it would probably be able to provide somewhat useful information and knowledge for utilizing walnut products as health food. Therefore, the manuscript might be worth.

Generally spesking, there were grammatical mistakes observed in this manuscript, and therefore should be more careful about the use of articles and the select of singular and plural forms. In particular, the wording seemed quite unique, or even not common used in the manuscript. It might be considered to be not helpful to read and understand the manuscript.

Round 2

Reviewer 1 Report

The revised manuscript has improved compared to the original version. The authors tried to address my questions as much as possible. I recommend the manuscript to be published!

Overall, the English language and grammar quality are good and acceptable.
